# Influences of the Contact State between Friction Pairs on the Thermodynamic Characteristics of a Multi-Disc Clutch

**DOI:** 10.3390/ma15217758

**Published:** 2022-11-03

**Authors:** Liang Yu, Changsong Zheng, Liyong Wang, Jianpeng Wu, Ran Jia

**Affiliations:** 1Ministry of Education Key Laboratory of Modern Measurement and Control Technology, Beijing Information Science and Technology University, Beijing 100192, China; 2School of Mechanical Engineering, Beijing Institute of Technology, Beijing 100081, China

**Keywords:** multi-disc clutch, contact state, oil film, friction torque, surface temperature

## Abstract

The relationship between clutch thermodynamic characteristics and contact states of friction components is explored numerically and experimentally. The clutch thermodynamic numerical model is developed with consideration of the contact state and oil film between friction pairs. The clutch bench test is conducted to verify the variation of the clutch thermodynamic characteristics from the uniform contact (UCS) to the intermittent contact (ICS). The results show that the oil film decreases gradually with increasing temperature; the lubrication state finally changes from hydrodynamic lubrication to dry friction, where the friction coefficient shows an increasing trend before a decrease. Thus, the friction torque in UCS gradually increases after the applied pressure stabilizes. When the contact state changes to ICS, the contact pressure increases suddenly and the oil film decreases rapidly in the local contact area, bringing about a sharp increase in friction torque; subsequently, the circumferential and radial temperature differences of friction components expand dramatically. However, if the contact zone is already in the dry friction state, friction torque declines directly, resulting in clutch failure. The conclusions can potentially be used for online monitoring and fault diagnosis of the clutch.

## 1. Introduction

The wet multi-disc clutch is one of the most important components in the automobile transmission system, which can smoothly engage and temporarily separate the active part and passive part of the transmission [1,2]. During the clutch engagement process, the friction components should transmit a large friction torque and bear a great thermal load, which can lead to buckling failure and local contact between friction pairs [3]. If clutch failure is not detected in time, deformation or wear of the friction components will be further aggravated in the subsequent working process, resulting in serious safety hazards [4].

Considerable research efforts have been devoted to the clutch failure mechanism [5,6]. Audebert N et al. [7] found that thermal buckling was related to the geometric shape and material properties of friction plates. Xiong C et al. [8] verified the applicability of Timoshenko Theory in the thermal bucking of friction components; when the circumferential bending moment caused by circumferential thermal stress exceeded the critical bending moment, the thermal buckling of friction components occurred inevitably. Li M et al. [9] developed a critical buckling moment model of friction components, suggesting that mechanical buckling and local high pressure were more likely to occur under a large mechanical torque. Moreover, Abdullah O I et al. [10] pointed out that thermo-elastic behavior was one of the main reasons for the premature failure of friction components; they also indicated that inhomogeneity of the pressure distribution was responsible for the increase in local thermal stress [11]. Zhao E et al. [12] proposed a temperature model of friction components under the local contact state, showing that concentrated contact pressure could bring about a significant temperature gradient. Yu L et al. [13] investigated the failure mechanism and position of vulnerable friction components in a multi-disc clutch and proposed the thermo-mechanical safety boundary.

However, researchers have consistently ignored how clutch failure is accompanied by variation of the contact and lubrication states between friction pairs [14]. As shown in Figure 1, a deformed plate can bring about local or intermittent contact between friction pairs, resulting in a high-load zone [15]. Thus, a local temperature gradient appears and is regarded as a hotspot; subsequently, serious wear zones show up, which are bright areas [16]. Moreover, automatic transmission fluid (ATF) plays an important role in identifying the contact state of friction pairs [17,18]. At the beginning of clutch engagement, the ATF is filled on a mating surface and viscous torque is dominant, which is called the hydrodynamic lubrication stage [19]. Subsequently, under the action of applied pressure, ATF is squeezed out of the clutch or into the pores of the friction disc, resulting in a decrease in the ATF film. As the asperities of the friction surface begin to make contact, a mixed lubrication state is entered [20]. At the end of clutch engagement, asperity contact is dominant, and a large amount of friction heat is generated; the friction pair enters the boundary lubrication state and, in severe conditions, even the dry friction state [21].

Typically, the Reynolds equation is used to predict variations of the ATF film and friction torque between friction pairs [22]. Kong J and Jang S [23] investigated convective cooling behaviors and temperature distributions of friction components. Bao et al. [24] found that the axial movement and surface morphology of friction pairs had a vital influence on the clutch friction torque. Cui et al. [25] investigated the dynamic characteristics of ATF film between deformed plates, indicating that both axial and radial deformations could result in an increase in viscous torque. At present, the macroscopic uniform contact model is often adopted in clutch thermodynamic investigations, and local contact caused by plate deformation is overlooked. There is an urgent need to rectify this research imbalance, but it is still a significant challenge to study the influence of the contact state of friction pairs on clutch lubrication and thermodynamic characteristics.

By considering the contact state and oil film between friction pairs, this paper proposes a numerical model to study clutch thermodynamic characteristics from the uniform contact state (UCS) to the intermittent contact state (ICS). Additionally, a clutch bench test is conducted to verify the simulation results. The conclusions provide a faithful theoretical basis for online monitoring, state evaluation and fault diagnosis of the clutch.

## 2. Numerical Model

### 2.1. COF Model

The lubrication state between friction pairs is related to the variation of ATF film. Thus, the COF model is developed considering ATF film loss, which is related to both COF in the dry friction state (*μ_a_*) and of the boundary film (*μ_b_*) [26]:(1)$$\mu =\alpha \left(T,u\right)\mu_{a}+\left[1-\alpha \left(T,u\right)\right]\mu_{b}$$
where α is the relative oil film loss; *T* and *u* represent the temperature and linear velocity of friction components, respectively.

Considering the lubrication and contact status, the proportion of boundary film is:
(2)$$1-\alpha =e^{-\frac{t_{l}}{t_{r}}}$$
where *t*_l_ is the time that the friction surface takes to pass through the contact length *l* with the velocity *u*; *t*_r_ is the average time that the absorbed molecules occupy the contact surface.
(3)$$t_{r}=t_{0}\cdot e^{\left(\frac{\varepsilon }{RT}\right)}$$
(4)$$t_{0}=4.75\times 10^{-13}(\frac{M_{\mathrm{mol}}V_{\mathrm{mol}}^{\frac{2}{3}}}{0.4T_{\mathrm{cr}}}{)}^{\frac{1}{2}}$$
(5)$$l=1.46\times 10^{-8}V_{\mathrm{mol}}^{\frac{1}{3}}$$where *R* is the gas constant and *ε* represents the adsorption heat; *M*_mol_ and *V*_mol_ are the molar mass and volume of ATF, respectively; *T*_cr_ is the critical temperature of ATF.

Integrating Equations (1)–(5), the proportion of the boundary film can be derived as [27]:(6)$$1-\alpha =\frac{\mu_{a}-\mu }{\mu_{a}-\mu_{b}}=\exp\left\{-\exp\left\{-\frac{\varepsilon }{RT}-\ln\left[5.15\times 10^{-5}u{\left(\frac{M_{\mathrm{mol}}}{T_{\mathrm{cr}}}\right)}^{\frac{1}{2}}\right]\right\}\right\}$$

A large number of experimental and simulation studies show that *μ_a_* is a function of temperature, whereas *μ_b_* is a function of temperature and pressure. Thus, Equation (1) can be modified as:
(7)$$\mu \left(T,u,p\right)=\alpha \left(T,u\right)\mu_{a}\left(T\right)+\left[1-\alpha \left(T,u\right)\right]\mu_{b}\left(T,p\right)$$


Considering the thermal decay of Cu-based friction material, the formula of *μ_a_* is obtained by Fourier function fitting:
(8)$$\begin{array}{ll} \mu_{a}(T)= & 0.196-0.088\cos(0.008T)+0.152\sin(0.008T)\\ & -0.005\cos(0.016T)+0.054\sin(0.016T) \end{array}$$


The type of ATF is RP-4652D and the parameters are listed in Table 1. Considering the viscosity of ATF and the porous diffusion of friction material, *μ_b_* is given as [28]:
(9)$$\mu_{b}(T,p)=0.04\times \exp\left[0.85\times {\mathrm{10}}^{-8}p-0.03(T-T_{e})\right]$$


### 2.2. Thermodynamic Model

When the separate plate is deformed, the contact state between friction pairs changes from a UCS to an ICS, as illustrated in Figure 2. Notably, contact pressure in the UCS is evenly distributed on the mating surface, whereas contact pressure in the ICS is concentrated in the local contact zone.

The three-dimensional heat-conduction equation of the friction components can be presented as follows [13]:
(10)$$\rho c\frac{\partial T}{\partial t}=\frac{1}{r}\frac{\partial }{\partial r}\left(\lambda r\frac{\partial T}{\partial r}\right)+\frac{1}{r^{2}}\frac{\partial }{\partial \theta }\left(\lambda \frac{\partial T}{\partial \theta }\right)+\frac{\partial }{\partial z}\left(\lambda \frac{\partial T}{\partial z}\right)$$


When the mating surface is in the UCS, contact pressure remains uniform in the circumferential direction (*∂p*/*∂θ*= 0); thus, heat transfer only occurs in the radial and axial directions. When the mating surface is in the ICS, the contact period of the separate plate is consistent with the number of spline teeth [29]. Moreover, the contact zone is designed as a sector to facilitate mesh generation. As the heat conduction process is the same for each contact period, the simulation calculation zone can be simplified as one contact period.

Based on the Equation (7), COF is related to surface temperature, velocity and contact pressure. Therefore, the COFs of different radial and circumferential positions are also different. The heat flux generated on the mating surface can be given as [29]:
(11)q(r,θ)=μ(r,θ)⋅p(r,θ)⋅ω⋅r


Since friction heat is conducted proportionally to the friction components, the heat partition factor can be expressed as:
(12)$$\gamma =\frac{\sqrt{\lambda_{s}\rho_{s}c_{s}}}{\sqrt{\lambda_{s}\rho_{s}c_{s}}+\sqrt{\lambda_{f}\rho_{f}c_{f}}}$$


Accordingly, heat flux conducted into the plate and disc can be given as:
(13)$$\left\{ \begin{array}{l} q_{s}=\gamma q\\ q_{f}=\left(1-\gamma \right)q \end{array}\right.$$


Contact pressure is equal to the applied pressure (*p*_app_) in the UCS. As for the ICS, the contact pressure in one period can be given as:
(14)$$p\left(r,\theta \right)=\{\begin{array}{ll} \frac{p_{\mathrm{app}}\cdot nS_{ABCD}}{nS_{abcd}}=\frac{p_{\mathrm{app}}}{\eta }, & \left(\mathrm{Contact zone}\right)\\ 0, & \left(\mathrm{Non-contact zone}\right) \end{array}$$
where *η* is the contact ratio of the mating surface.

Boundary conditions in the UCS are as follows:
(15)$$\left\{ \begin{array}{l} {\lambda \frac{\partial T\left(r,z,t\right)}{\partial r}|}_{r=r_{i}{\mathrm{, CC}}_{z}}=+\kappa_{i}\left[T\left(r,z,t\right)-T_{e}\right],\;0\leq z\leq h\\ {\lambda \frac{\partial T\left(r,z,t\right)}{\partial r}|}_{r=r_{o}{\mathrm{, BB}}_{z}}=-\kappa_{o}\left[T\left(r,z,t\right)-T_{e}\right],\;0\leq z\leq h\\ {\lambda \frac{\partial T\left(r,z,t\right)}{\partial z}|}_{z=0\mathrm{, BC}}=q_{s,i}\left(r,\theta ,t\right),\;r_{i}\leq r\leq r_{o}\\ {\lambda \frac{\partial T\left(r,z,t\right)}{\partial z}|}_{z=h{\mathrm{, B}}_{z}C_{z}}=q_{s,i+1}\left(r,\theta ,t\right),\;r_{i}\leq r\leq r_{o} \end{array}\right.$$


In addition to the above boundary conditions, the mating surface in the ICS contains two insulated boundaries (AD and BC) as follows:
(16)$$\left\{ \begin{array}{l} {\lambda \frac{\partial T\left(r,\theta ,t\right)}{\partial \theta }|}_{\mathrm{AD}}=0,\;r_{i}\leq r\leq r_{o},\;0\leq z\leq h\\ {\lambda \frac{\partial T\left(r,\theta ,t\right)}{\partial \theta }|}_{\mathrm{BC}}=0,\;r_{i}\leq r\leq r_{o},0\leq z\leq h \end{array}\right.$$


Conclusively, the friction torques in the UCS and ICS can be given as:
(17)$$M_{\mathrm{UCS}}=\int_{r_{i}}^{r_{o}}\int_{0}^{2\pi }r^{2}\frac{p_{\mathrm{app}}S\cdot \mu }{S}drd\theta =\frac{2\pi p_{\mathrm{app}}\left(r_{o}^{3}-r_{i}^{3}\right)}{3}\int_{r_{i}}^{r_{o}}\mu \;dr$$
(18)$$M_{\mathrm{ICS}}=n\int_{r_{i}}^{r_{o}}\int_{0}^{\frac{2\pi }{n}}r^{2}\mu \frac{p_{\mathrm{app}}}{\eta }drd\theta =\frac{n\cdot p_{\mathrm{app}}\left(r_{o}^{3}-r_{i}^{3}\right)}{3\eta }\int_{r_{i}}^{r_{o}}\int_{0}^{\frac{2\pi }{n}}\mu drd\theta$$


## 3. Experimental Method

An experimental investigation was conducted to verify the clutch thermodynamic characteristics from the UCS to the ICS. The clutch test bench was clearly introduced in our previous studies [13], mainly including the driveline, hydraulic system and measurement and control system. In the driveline module, the motor drives the active end of the clutch to rotate, while the passive end is locked by the brake drum. As for the hydraulic system, it provides ATF for all clutch components and absorbs friction heat; clutch engagement and disengagement are controlled by electro-hydraulic proportional valves. As shown in Figure 3, the clutch is a six-friction-pair system, and the material and dimensional parameters of friction components are listed in Table 2. The thermometric plate undergoes circumferential elastic deformation before the test; it is arranged in the axial middle position of the clutch so that both friction surfaces have heat flux input to increase the probability of dry friction. Moreover, five thermometer holes are evenly spaced between the two teeth of the plate to measure the temperature at different radial positions. The surface temperature is measured by a K-typed thermocouple, where the response time does not exceed 20 ms and the measurement error is within ±2 °C.

The experimental conditions are listed in Table 3. As the experiment is a destructive test of friction components, the number of repeated tests in each working condition is different for the safety of testers and equipment. Moreover, the temperature data from the second test are selected in all repeated experiments. In order not to aggravate the damage to the deformed plate and to clearly observe its thermodynamic characteristics, the applied pressure and rotating speed of the clutch are low with a long sliding time. As the input heat flux is different in each test, the sliding time is also different. Local contact can cause a high surface temperature, leading to carbonization and gasification of ATF. Therefore, if the bench fumes, the experiments must be stopped immediately by unloading the piston pressure and turning off the motor.

## 4. Results and Discussion

### 4.1. Simulation Results

#### 4.1.1. COF

Figure 4 shows the influence of surface temperature on COF under an applied pressure of 1.5 MPa. As shown in Figure 4a, the variation of COF can be divided into three stages under 1500 rpm. As the high speed promotes the hydrodynamic effect between friction pairs, the ATF film plays a dominant role in supporting the applied pressure in stage I. Since an increase in temperature can reduce the viscosity of ATF, the ATF film decreases, and asperity contact gradually appears. However, the decline of COF caused by the decrease in ATF viscosity is much greater than the increase in COF caused by the ATF film loss. Therefore, the COF sees a decreasing trend with the increasing temperature. 

As the temperature increases in stage II, the contact state of the friction pair changes from mixed lubrication to boundary lubrication, and finally enters the dry friction state, which is consistent with the Stribeck curve. Meanwhile, the thickness of the ATF film gradually reaches the minimum, which is much smaller than the height of asperities [30]. Accordingly, asperity contact is dominant, which does not simply bring about a dramatic increase in COF but also promotes the decomposition and oxidation of ATF. When the friction pair is in the dry friction state, COF reaches the highest value at about 0.35 and is temporarily stable. Subsequently, thermal decay occurs and becomes increasingly worse in stage III, leading to a significant decrease in COF. Moreover, as the friction material begins to soften, adhesive friction plays a leading role and is accompanied by the peeling of friction material. Therefore, not only is the wear of friction material apparently accelerated but the clutch service life is also significantly shortened. Such a phenomenon has been experimentally confirmed via a metallographic micrograph of friction material [31].

When the rotating speed is relatively low, the ATF film will be more likely to be squeezed into the pores of a friction disc or flow away. Thus, the ATF film decreases, and the hydrodynamic effect weakens, leading to the absence of stage I. As illustrated in Figure 4b, the friction surface directly enters the mixed lubrication state, and thus COF increases parabolically, reaching the highest value of 0.354 at 180 °C. Consequently, the COF of a friction pair with a lower rotating speed is higher at the same temperature.

#### 4.1.2. Thermodynamic Properties

The UCS and ICS were considered to study the clutch thermodynamic properties. For the former, the simulation conditions were the same as tests 1 and 5; the latter was consistent with test 5. There were 18 uniformly distributed contact zones in the ICS, corresponding to the number of spline teeth. The contact ratio was 0.375; the inner and outer radii of the contact sector were 95 mm and 115 mm, respectively, and the central angle was 15°. Moreover, the applied pressure increased and reached a maximum value at 3 s, then remained constant.

As shown in Figure 5a,b, there are no circumferential temperature differences in the UCS. The surface temperature increases along the radial direction, resulting in a gradual increase in the radial temperature difference. The highest temperature in the sliding time rises to 77.74 °C in Figure 5a, and the radial temperature difference extends from zero to 20.45 °C. As demonstrated in Figure 5c, the surface temperature in the contact sector is far greater than that in the non-contact zone, and thus a large temperate gradient shows up in both the radial and circumferential directions; the radial temperature shows a convex parabolic increase, whereas the circumferential temperature changes sinusoidally. Although the working parameters in Figure 5b,c are consistent, the contact pressure in the ICS is larger than that in the UCS, which leads to a higher heat flux. Thus, their highest temperatures are 143.36 °C and 310.80 °C, respectively.

As shown in Figure 6, the friction torque increases linearly with the increase in applied pressure at first. When the applied pressure stabilizes, the friction torque continues to increase with the change in temperature and ATF film. More specifically, the friction torque in Figure 6a increases from 146.94 N·m to 206.48 N·m. Compared with the UCS in Figure 6b, the local contact pressure in the ICS is much higher in Figure 6c. Therefore, more heat flux is generated in the local contact area and the ATF film loss is aggravated, contributing to the rapid growth of friction torque. The friction torque reaches the peak value of 460 N·m at 5.4 s. According to Figure 5c, since the surface temperature has reached 177 °C, the friction surface gradually enters the dry friction state. Therefore, thermal decay of friction material appears, accompanied by a reduction in friction torque.

### 4.2. Experimental Results

In the repeated experiments, the variation of friction torque was highly consistent. A maximum relative error of friction torque of only 6.8% appeared in test 4, thus confirming the reproducibility of the experiment. During the clutch sliding period, the variations of friction torque could be classified into three categories: linear, step and convex parabolic. As shown in Figure 7a,c, when the heat flux is small, the friction torque increases almost linearly after the applied pressure is stable; the larger the applied pressure, the greater the growth rate of friction torque, which is consistent with the simulation results. Exemplifying this, the growth rates are 2.57 N·m/s and 7.22 N·m/s, respectively. At the end of the sliding time, the radial temperature differences of both conditions are 24.54 °C and 47.35 °C, respectively, and the circumferential temperature differences are quite small, namely, 0.81 °C and 1.18 °C. Since the radial and circumferential temperature differences are within the safe range, the thermometric plate is still in the UCS.

Moreover, the simulation and experimental results can be quantitatively compared before the plate deformation. As shown in Figure 6a and Figure 7a, the largest friction torques are 206.48 N·m and 203.26 N·m, respectively. Additionally, the variation of temperatures under the simulation and test are also consistent, as presented in Figure 5a and Figure 7a, where the largest temperatures in the different radial directions are 77.74 °C, 70.16 °C and 57.29 °C and 77.73 °C, 64.97 °C and 53.19 °C, respectively. The maximum relative error of the thermodynamic parameters is less than 8%. Therefore, the numerical model has been verified by the experimental results, which offer an effective evaluation of the clutch thermodynamic performance.

As shown in Figure 7b,d, when changing the rotating speed to increase the heat flux, the friction torque changes remarkably and can be divided into three stages. The variation of friction torque in stage I is the same as in Figure 7a,c. However, stage II is a transition stage from the UCS to the ICS, where the friction torque rises sharply. For example, the friction torque increases by 12.9 N·m in 0.69 s in Figure 7d. Moreover, there is a time turning point *t_d_* after which the circumferential temperature differences gradually arise; the differences expand to 3.68 °C and 9.16 °C at the end of the sliding time, respectively. As for stage III, the friction torque stabilizes temperately and then shows a downward trend. However, the test bench emitted oil fumes under test 4 so the experiment had to be stopped at 9 s. This phenomenon strongly proves that the ICS leads to a decline of ATF film, which then induces dry friction.

As the heat flux continues to increase in Figure 7e,f, the variation of friction torque is consistent with that of COF in Figure 4b, which can be divided into two stages, namely, the increase stage I and the decrease stage II. In stage I, the friction torques increase dramatically in both conditions, even after the applied pressure stabilizes; the friction torques peak at *t*_1_, where the values are 426.3 N·m and 410.8 N·m, respectively. Moreover, as the highest temperatures at *t*_1_ are 150.5 °C and 177.4 °C, respectively, the contact zone has already entered the dry friction state. Subsequently, the friction torque is continuously attenuated. Since the real contact area and position of the thermometric plate are uncertain in experiments, the simulation and experimental results in the ICS can only be qualitatively compared. When the contact ratio is set as 0.375 in the simulation, the friction torque reaches the highest value of 460 N·m and the surface temperature is 177 °C; then, the friction torque shows significant attenuation. Both the simulation and experiment show that ATF film reduces rapidly in the ICS, resulting in thermal decay of friction material and attenuation of friction torque. The above findings will be applied to clutch online monitoring.

The radial temperature differences in Figure 7e,f are 101.69 °C and 120.10 °C at *t*_1_, respectively, suggesting that radial thermal buckling has occurred. There is also a turning point *t_d_* after which the circumferential temperature differences arise. It should be noted that ATF film loss can contribute to local contact, bringing about further expansion of the temperature difference. Consequently, circumferential deformation begins after *t_d_*; since the applied pressure has not reached the predefined value at this time, friction torque continues to rise, rather than showing a step increase. Therefore, the plate has been deformed both radially and circumferentially at *t*_1_. Finally, at the end of the sliding time, the largest circumferential temperature differences in the two tests are 9.31 °C and 20.94 °C, respectively, and the radial temperature differences also increase to 130.56 °C and 165.48 °C, respectively. Meanwhile, vaporized ATF fumes are produced in both conditions. There is no denying that the clutch has failed, and the separate plate has undergone severe and comprehensive deformation.

## 5. Conclusions

Based on the structural characteristics and failure form of the friction components in a Cu-based wet clutch, a comprehensive thermodynamic numerical model, as well as a COF model, have been proposed considering the effect of the ATF film between friction pairs. The influence of the contact state of friction components on thermodynamic characteristics of the clutch has been studied and verified numerically and experimentally. The conclusions shed light on online monitoring and state evaluation of the clutch, and may be summarized as follows:When the friction pairs are in the UCS, an increase in temperature causes the ATF film to gradually decrease, and thus friction torque shows a progressive increase trend;When the contact state changes from the UCS to the ICS, the local contact pressure and COF increase significantly, bringing about a step increase in friction torque. Subsequently, the circumferential and radial temperature differences of friction components expand dramatically;As the surface temperature increases in the ICS, the ATF film becomes difficult to form and gradually decreases, resulting in dry friction. Therefore, friction torque decays directly, and the surface temperature differences increase rapidly, leading to clutch failure.

## Figures and Tables

**Figure 1 materials-15-07758-f001:**
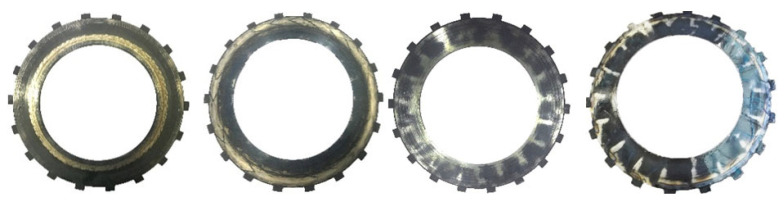
Wear of the deformed separate plates.

**Figure 2 materials-15-07758-f002:**
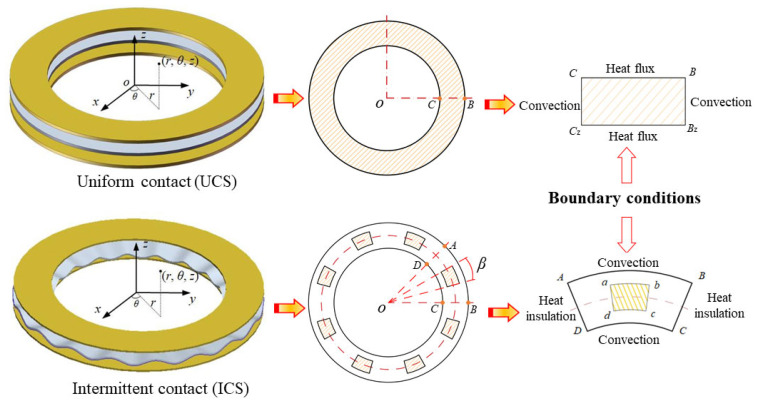
Boundary conditions of friction components.

**Figure 3 materials-15-07758-f003:**
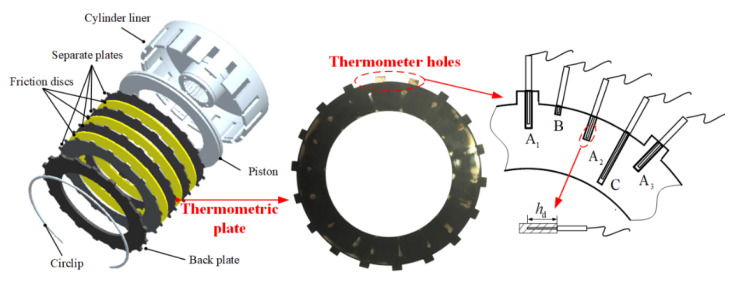
Temperature measurement of the clutch.

**Figure 4 materials-15-07758-f004:**
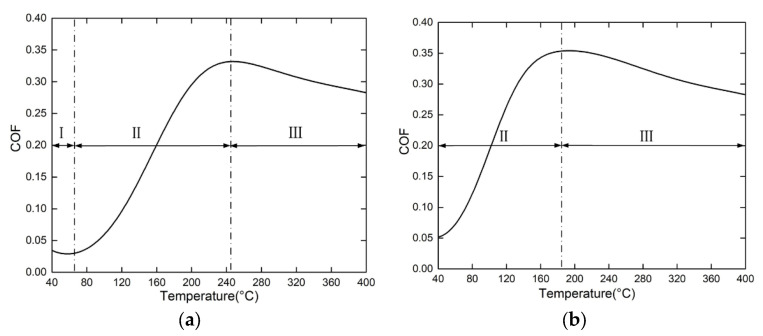
Influence of surface temperature on COF. (**a**) 1500 rpm; (**b**) 300 rpm.

**Figure 5 materials-15-07758-f005:**
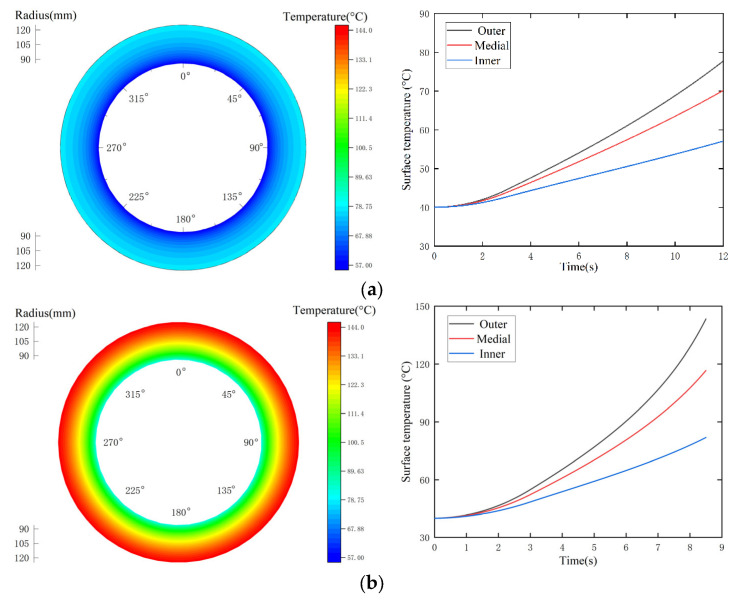
Simulation results for the surface temperature. (**a**) Test 1 (UCS); (**b**) Test 5 (UCS); (**c**) Test 5 (ICS).

**Figure 6 materials-15-07758-f006:**
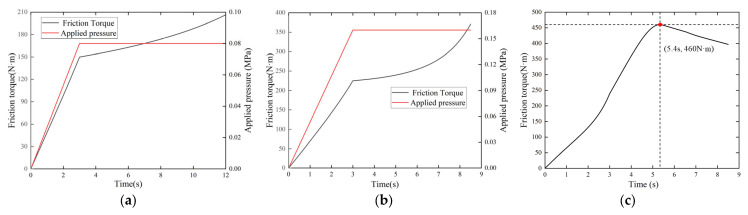
Simulation results for the friction torque. (**a**) Test 1 (UCS); (**b**) Test 5 (UCS); (**c**) Test 5 (ICS).

**Figure 7 materials-15-07758-f007:**
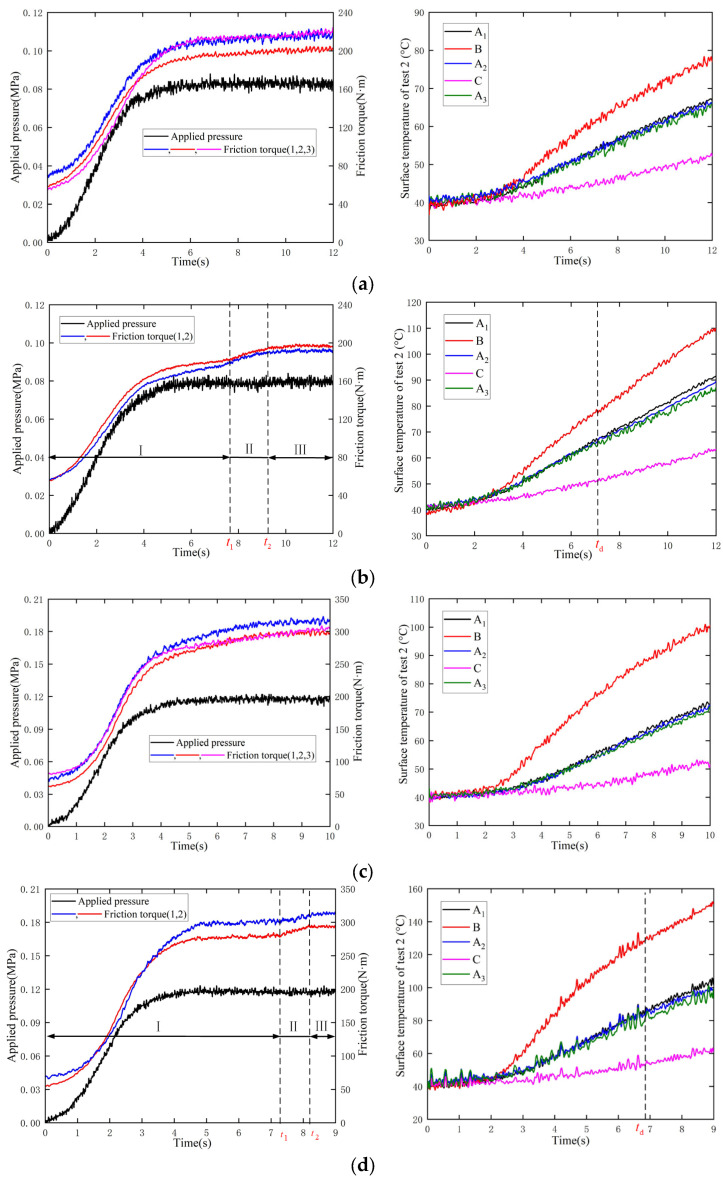
Experimental results for the clutch thermodynamic characteristics. (**a**) Test 1; (**b**) Test 2; (**c**) Test 3; (**d**) Test 4; (**e**) Test 5; (**f**) Test 6.

**Table 1 materials-15-07758-t001:** ATF parameters (RP-4652D).

Parameters	Value
Viscosity grade	SAE 15 W–40
Flash point	255 °C
Molar mass (*M*_mol_)	310 g/mol
Gas constant (*R*)	8.3
Critical temperature (*T*_cr_)	353.15 K
Adsorption heat (*ε*)	31,425 J/mol

**Table 2 materials-15-07758-t002:** Thermophysical parameters of the wet clutch.

Variable	Friction Core	Friction Lining	Separate Plate
Density, kg/m^3^	7800	5500	7800
Specific heat, J/(kg·°C)	487	460	487
Thermal conductivity, W/(m·°C)	45.9	9.3	45.9
Radius, mm	86/125	86/125	86/125

**Table 3 materials-15-07758-t003:** Experimental conditions.

Test	1	2	3	4	5	6
Applied pressure (MPa)	0.08	0.08	0.12	0.12	0.16	0.16
Rotating speed (rpm)	150	300	150	300	300	600
Sliding time (s)	12	12	10	9	8.5	5.96
Repeated times	3	2	3	2	1	1

## Data Availability

Not applicable.

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
