# Peer review of "Influences of the Contact State between Friction Pairs on the Thermodynamic Characteristics of a Multi-Disc Clutch"

_materials, 2022, doi:10.3390/ma15217758_

Round 1
Reviewer 1 Report
The influence of the contact state of friction components on the thermodynamic characteristics of the clutch has been studied and verified numerically and experimentally in this manuscript.
The paper is sufficiently organized; however, the section with the nomenclature and abbreviations should be added, then all the variables and shorts will be in one place, which helps the potential readers to understand the text, and, it will not be necessary to browsing of the entire paper, to find the meaning of the variables and shortcuts. Additionally, it allows for avoiding situations such as using the same tag to describe different variables.
In addition to numerical analysis, the authors also realize experiments that increase the quality of the paper. However, the paper contains limited discussions, especially since there is a lack of comparison to the results of other authors, therefore a significant discussion regarding the underlying mechanisms controlling the observed findings should be realized.
In addition, the authors should take into consideration the following things:
How does the axis system adopted in the figures correspond to the symbols adopted in the equations? - Please be consistent in this matter.
Page 7 line 198 - it is written: ‘Meanwhile, the ATF film thickness gradually reaches the minimum which is much smaller than the height of asperities’ Where in the paper there is information on the height of the asperities. Where in the model do the asperities take into consideration?
In Figure 2, the ‘x’ is used to describe the x-axis of the coordinate system. Explain what is the meaning of 'x' in Equation 5.
The quality of figure 6 should be improved especially the description of the particular elements should have better quality (maybe increasing the font size will be enough).
Reviewer 2 Report
The article submitted for review has a high scientific and practical value. It discusses the issues of modeling, taking into account experimental data, friction pairs in the clutch of cars. Both the mathematical model and the results of a series of field experiments are of interest. However, the work is not without drawbacks:
1) The description of the mathematical model in the second section of the article suffers from the lack of many explanations. So in equations (10) and (11), which are given after Figure 2, the distribution of the temperature field is represented in cylindrical coordinates, which are not disclosed by the authors of the work. The coordinates presented by the authors in Figure 2 are Cartesian coordinates. The authors should disclose the cylindrical coordinate system they use in Figure 2.
2) In expressions (14), (15), (16) new constants and variables are introduced into the reasoning, such as: ki, k0, h, ri, r0, etc., which are not disclosed by the authors in the explanations to the expressions. In this regard, the authors need to prepare a list of used symbols and abbreviations for the article, which will significantly improve the quality of the work.
3) From the 2nd section of the article, it is not clear what the authors personally did in the mathematical model. This needs to be voiced either in this section or in the general conclusion. The author's contribution to the mathematical modeling of friction processes is an interpretation of the scientific level of the article.
4) In the Introduction, the authors already summarize the results of the entire work, claiming that a new model of the coefficient of friction has already been developed. It is more correct, from the point of view of the interpretation of the research results, to formulate a scientific task in the Introduction. The motivation for achieving this task should be presented in the conclusion.
5) The bibliographic list consists of 75% of publications by authors from the PRC, including those with a large volume of self-citation, which shows a certain degree of bias in the selection of sources. It is required either to revise the bibliographic list, or to expand the list by adding publications of authors from other countries to it.
Reviewer 3 Report
Authors of the publication „Influences of the contact state between friction pairs on the thermodynamic characteristics of a multi-disc clutch” presented the results numerical calculation and research experimentally the relationship between the clutch (wet multi-disc clutch) thermodynamic characteristics and the contact states of friction components.
Each of the presented parts of the publication has been correctly described by the authors. The conclusions are consistent and closely related to the research topic. As a reviewer of this work, however, I believe that the reviewed work requires small corrections, which will undoubtedly improve its quality.
1. Introduction - authors cite publications from one country. Maybe it is worth reading the research of authors from other countries.
2. Line 159 - I think that a few words about the clutch test bench should be written.
3. Figure 7. Experimental results of the clutch thermodynamic characteristics - It is worth adding the designation A1, B, A2, C, A3 as shown in the figure 3.
Each of the graphs is presented in a very legible and clear way, as well as their interpretation is described in detail. The work presented for review is very interesting, deals with an interesting topic and fits well with the scope of the journal. The research methodology presented is good. The conclusions are rightly worded. The research results and their analysis are also adequately presented. Summing up, the reviewed work presents a very high substantive and experimental value.
Round 2
Reviewer 1 Report
|
The authors have included every comment and suggestion. The paper is suitable for publication. |